# Graph Convolutional Networks for Cervical Cell Classification

Jun Shi[1], Ruoyu Wang[1], Yushan Zheng[2,3,4], Zhiguo Jiang[2,3,4] and Lanlan Yu[5]

[1] School of Software, Hefei University of Technology, Hefei 230601, China
`juns@hfut.edu.cn, aywry@mail.hfut.edu.cn`
[2] Image Processing Center, School of Astronautics, Beihang University, Beijing, 100191, China
[3] Beijing Advanced Innovation Center for Biomedical Engineering, Beihang University, Beijing, 100191, China
[4] Beijing Key Laboratory of Digital Media, Beihang University, Beijing, 100191, China
`{yszheng, jiangzg}@buaa.edu.cn`
[5] Motic (Xiamen) Medical Diagnostic Systems Co. Ltd., Xiamen 361101, China
`yull@motic.com`

**Abstract.** Cervical cell classification is of important clinical significance in the screening of cervical cancer at early stages. In this paper, we present a novel cervical cell classification method based on Graph Convolutional Network (GCN). In contrast with Convolutional Neural Networks (CNN) which can classify cervical cells through learned deep features, the proposed method uses GCN to explore the image-level potential relationship for improving the classification performance. Specifically, each cervical cell image is represented by a pretrained CNN. $k$-means clustering is performed on these CNN features and then the graph structure is constructed where each node is characterized by one cluster centroid. Consequently, the image-level relationship can be captured in terms of intrinsic clustering structure. GCN is applied to propagate the underlying correlation of nodes and the relation-aware representation of GCN is incorporated to enrich the image-level CNN features. Experiments on the cervical cell image datasets demonstrate the effectiveness of our method.

**Keywords:** Cervical cancer, Cervical Cell Classification, Graph Convolutional Networks.

## 1 Introduction

Cervical cancer is one of the primary causes of cancer death in women [1]. Screening at early stages is of great importance to the prevention and early detection of cervical cancer. As the most popular screening test, cervical cytology has been widely used in many countries and effectively reduced the incidence and mortality. Currently manual screening of abnormal cells from a cervical cytology slide is still the common practice. However, it is generally tedious, inefficient and high-cost. Therefore, the automatic

screening method has gained increasing concern. It aims to automatically select abnormal cells from a given cytology slide, determine the categories and finally present the analysis results of the whole slide according to The Bethesda System (TBS).

Accurate cervical cell classification is crucial to the automatic screening method. Over the past decades, many cervical cell image classification methods have been developed. Chankong et al. [2] apply morphological features of cell to achieve the multi-label classification. Bora et al. [3] combine the shape, texture and color features of nuclei to classify the cervical dysplasia. These methods generally use handcrafted or engineered features and one or multiple classifiers for classification. Consequently, they are inevitably influenced by feature or classifier selection. In the last few years, Convolutional Neural Networks (CNN) have been proposed to automatically learn multi-level features through hierarchical deep architecture. A variety of CNN models have been successfully used in computational pathology [4, 5] and cervical cytology [6-9]. However, these methods ignore the potential relationship among images and thus may produce inaccurate feature representation.

In this paper, we propose a novel cervical cell classification method based on Graph Convolutional Network (GCN) [10]. Concretely all the CNN features of cervical cell images are clustered. The graph structure based on intrinsic clustering correlation is constructed where each node is represented by one cluster centroid. The corresponding adjacency relationship and node features are fed into GCN for learning relation-aware representation of nodes. The final GCN representation is incorporated to enrich the image-level CNN features. Experimental results on cervical cell image classification verify the feasibility and effectiveness of our method.

The contribution and novelty of this paper is two-folds. To the best of our knowledge, this is the first to apply GCN for cervical cell classification. The potential correlations of images are well preserved and the relation-aware representation generated by GCN greatly enhances the discriminant ability of CNN features. Moreover, the large-scale Motic liquid-based cytology image dataset with 7 categories is proposed. Each image is manually annotated by experts. The large amount of data and some novel cell types with important clinical significance provide a new challenge for cervical cell image analysis field.

## 2 Methodology

### 2.1 Overview

The classification framework of our method for cervical cell images is presented in Fig. 1. First a CNN model (e.g. DenseNet [11]) pre-trained by cervical cell classification task is used to extract features of each cervical cell image. $k$-means clustering is performed on these CNN features and thus the cluster centroids can be obtained. The graph of cluster centroid correlation is then constructed based on their intrinsic similarities. Afterwards, two-layer GCN is used to learn over the graph structure and node features, aiming to generate relation-aware representations of nodes. All of these encoded representations are further incorporated into the CNN features through dot product. Cross-entropy loss is applied to train the whole network after linear projection.

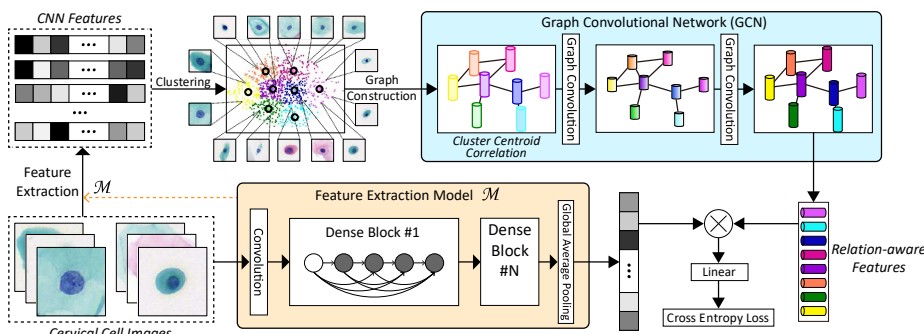

**Fig. 1.** Classification framework of our method for cervical cell images.

## 2.2    Graph Convolutional Network

The goal of Graph Convolutional Network (GCN) [10] is to learn the relation-aware representations of nodes via propagating the structure information of graph. Different from CNN which performs convolution on local Euclidean structure, GCN generalizes the operation of convolution to non-Euclidean data (e.g. graph). Particularly it usually operates convolution on features of neighbors for each node in the graph structure, and effectively combines the intrinsic structure information and node features in the learning process. Consequently, it can generate more informative representations for nodes. Given a graph $G = (V, E)$ with $N$ nodes $v_i \in V$, edges $(v_i, v_j) \in E$ and an adjacency matrix $A \in \mathbb{R}^{N \times N}$ which characterizes the correlation of nodes, GCN aims to encode the graph $G$ via a neural network model $f = (X, A)$ where $X \in \mathbb{R}^{N \times D}$ is the features of $N$ nodes. A multi-layer GCN updates the node features according to the following layer-wise propagation rule:

$$H^{(l+1)} = \sigma\big(\hat{A} H^{(l)} W^{(l)}\big) \tag{1}$$

$H^l \in \mathbb{R}^{N \times d_l}$ denotes the feature representations of nodes in the $l$th layer, $d_l$ indicates the feature dimension and $H^{(l+1)} \in \mathbb{R}^{N \times d_{l+1}}$ is the updated node features. Note that $H^{(0)} = X$. $\hat{A} \in \mathbb{R}^{N \times N}$ is the normalized version of the adjacency matrix $A$ [10]. $W^{(l)} \in \mathbb{R}^{d_l \times d_{l+1}}$ is a layer-specific trainable weight matrix and $\sigma(\cdot)$ denotes an activation function (e.g. ReLU). As a consequence, the relation-aware representations of nodes can be gained through a multi-layer GCN.

## 2.3    GCN for Cervical Cell Classification

To deal with cervical cell classification, CNN base model is firstly applied to represent each cervical cell image in this paper. In view of the potential correlation among images ignored in CNN learning process, GCN is introduced to explore the dependencies of images and simultaneously obtain the relation-aware representation. In contrast with conventional GCN [10] which applies the node-level representation for node classification, our method incorporates the GCN output into the deep CNN features. It could be regarded that the latent correlations of images are preserved in the final feature representations of cervical cell images. More importantly, the relation-aware property of

GCN greatly enhances the discriminant ability of CNN features and thus effectively improves the classification performance.

In this paper, DenseNet-121 architecture pre-trained by cervical cell classification task is used as the CNN base model. Global average pooling is applied to obtain the image-level feature representation $\boldsymbol{I} \in \mathbb{R}^D$ and the feature dimension $D = 1024$. $k$-means clustering is then performed on these CNN features and thus the cluster centroids $\boldsymbol{X} \in \mathbb{R}^{N \times D}$ can be obtained. Here $N$ denotes the number of cluster centroids. The intrinsic clustering correlation reflects the potential relationships of images to a great extent. Then the graph structure $G$ is constructed based on the cluster centroids $\boldsymbol{X}$ and the corresponding adjacency matrix $\boldsymbol{A} \in \mathbb{R}^{N \times N}$ is defined as follows:

$$\boldsymbol{A}_{ij} = \begin{cases} 1, & if \ \boldsymbol{X}_i \in KNN(\boldsymbol{X}_j) \ or \ \boldsymbol{X}_j \in KNN(\boldsymbol{X}_i) \\ 0, & otherwise \end{cases} \tag{2}$$

where $KNN(\boldsymbol{X}_i)$ denotes the $k$ nearest neighbors of $\boldsymbol{X}_i$ based on Cosine similarity. The adjacency matrix $\boldsymbol{A}$ and node features $\boldsymbol{X}$ are inputted into stacked two-layer GCN. According to the propagation rule of GCN shown in Eq. (1), the node features $\boldsymbol{H}^{(2)} \in \mathbb{R}^{N \times D}$ in the last layer can be learnt. Then, $\boldsymbol{H}^{(2)}$ is incorporated into the CNN features $\boldsymbol{I}$, and the relation-aware representation can be gained through dot product:

$$\boldsymbol{y} = \boldsymbol{H}^{(2)}\boldsymbol{I} \tag{3}$$

Finally, cross-entropy loss is used for the training of the whole network after linear projection $\boldsymbol{z} = \boldsymbol{y}\boldsymbol{W} + \boldsymbol{b}(\boldsymbol{z} \in \mathbb{R}^C)$, where $\boldsymbol{W} \in \mathbb{R}^{N \times C}$ is the weight, $\boldsymbol{b}$ is the bias and $C$ is the number of categories of cervical cells. At the inference time, the DenseNet features of each image are firstly gained and its corresponding GCN representation is then generated by pre-constructed graph and trained two-layer GCN. In the end, each image is represented by the combination of its DenseNet and GCN features.

## 3 Experiments

### 3.1 Datasets

To evaluate the performance of our method for cervical cell classification, SIPaKMeD [7] and Motic cervical cell image datasets are used in this paper. SIPaKMeD dataset contains 4049 images of isolated cells with 5 different categories from 966 cluster cell images of Pap smear slides shown in Fig. 2. Data distribution of SIPaKMeD is given in Table 1.

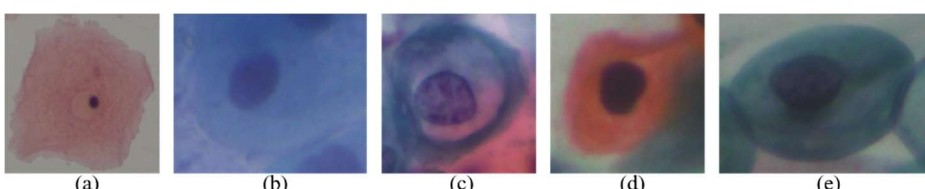

| (a) | (b) | (c) | (d) | (e) |

**Fig. 2.** Cell images of 5 categories in SIPaKMeD dataset: (a) Superficial-Intermediate, (b) Parabasal, (c) Koilocytotic, (d) Dyskeratotic, (e) Metaplastic.

**Table 1.** Data distribution of SIPaKMeD dataset.

| Category | Superficial/ Intermediate | Parabasal | Koilocytotic | Metaplastic | Dyskeratotic | Total |
|---|---|---|---|---|---|---|
| Num of Cells | 813 | 787 | 825 | 793 | 813 | 4049 |

In addition, the liquid-based cytology image dataset with 7 categories and 20× objective provided by Motic is used for cervical cytology research. It consists of 2 subsets, i.e., Subset-1 from 35 slides stained by Thinprep and Subset-2 from 111 slides stained by Motic medical laboratory. Cell images of Motic dataset are shown in Fig. 3 and the data distribution is listed in Table 2. Some novel cell types with important clinical significance (e.g. Granulocyte, Glandular cells, Koilocytotic cells and cells with high nuclear-cytoplasmic ratio) are included. It should be noted that each image is 128×128 pixels, which is collected in this way that the centroid of segmented nucleus in a slide is taken as the center and then the area 128×128 around the center is captured. The ground-truth of each cell is carefully annotated by expert.

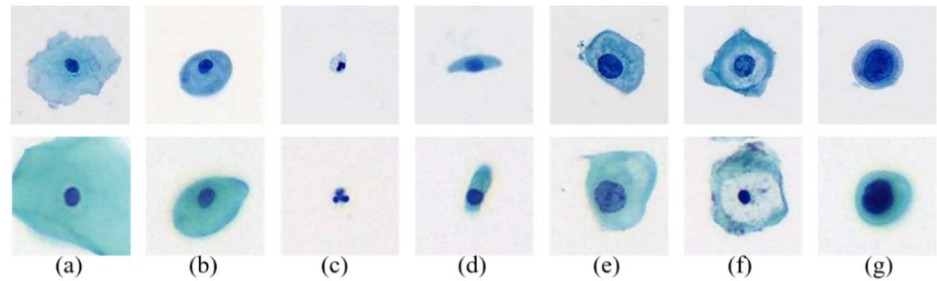

**Fig. 3.** Cell images of 7 categories in Motic dataset: (a) Superficial squamous cells, (b) Intermediate squamous cells, (c) Granulocyte, (d) Glandular cells, (e) Atypical squamous cells (Atypical), (f) Koilocytotic cells, (g) Cells with high nuclear-cytoplasmic ratio (High-N/C-Ratio). The first row is from Subset 1 and the last one is Subset 2.

**Table 2.** Data distribution of Motic dataset.

| Category | Num of Images of Subset-1 | Num of Images of Subset-2 | Property |
|---|---|---|---|
| Superficial | 2713 | 6872 | |
| Intermediate | 364 | 514 | Normal |
| Granulocyte | 5202 | 5527 | |
| Glandular | 36 | 596 | |
| Atypical | 764 | 1263 | |
| Koilocytotic | 382 | 151 | Abnormal |
| High-N/C-Ratio | 342 | 652 | |
| Total | 9803 | 15575 | 25378 |

## 3.2 Experimental Settings

Each cervical cell image is represented by a 1024-dimensonal DenseNet-121 feature vector. The number of $k$-means clustering is selected from {128, 256, 512, 1024, 2048}. The neighbor parameter is tuned from 10 to 100 at intervals of 10. The output dimension of the first GCN layer varies from {256, 512, 1024, 2048} and is determined as 256. The final GCN dimension is 1024 due to the incorporation of DenseNet-121 and GCN features. Following the practice [7, 8], 5-fold cross-validation is applied for evaluation of classification performance.

Our method is compared with ResNet-101[12], DenseNet-121 and CNN features proposed in SIPaKMeD dataset [7]. Particularly the 27-dimensional cell morphological features (Morphological-27), including 26-dimensional cell features [7] and the perimeter of the nucleus, are used for Motic dataset, and SVM with Radial Basis Function (RBF) kernel serves for the classifier. Our model is implemented on PyTorch. All the experiments are performed on a computer with an Intel Core i7-7820X CPU of 3.60 GHz and a GPU of NVIDIA GTX 1080Ti.

## 3.3 Results

**Results on SIPaKMeD dataset.** Following the practice [7], as the input to the network, each cropped cell images of SIPaKMeD is resized to 80×80 pixels. The number of $k$-means clustering and the neighbor parameter are respectively set as 1024 and 40 by experiments. The classification results are presented in Table 3. As shown, our method yields better classification performance compared with other methods. It can be explained that the potential correlations of images are well preserved in the relation-aware representation generated by GCN, which contributes to more discriminative ability. Confusion matrices of classification are exhibited in Fig. 4. As the most challenging cells, the classification results of Koilocytotic cells in our method achieve 97.45%.

**Table 3.** Comparison of classification accuracies (mean±std) on SIPaKMeD dataset (%).

| Methods | CNN features [7] | ResNet-101 | DenseNet-121 | Our method |
|---------|------------------|------------|--------------|------------|
| Accuracy | 95.35±0.42 | 94.86±0.74 | 96.79±0.42 | 98.37±0.57 |

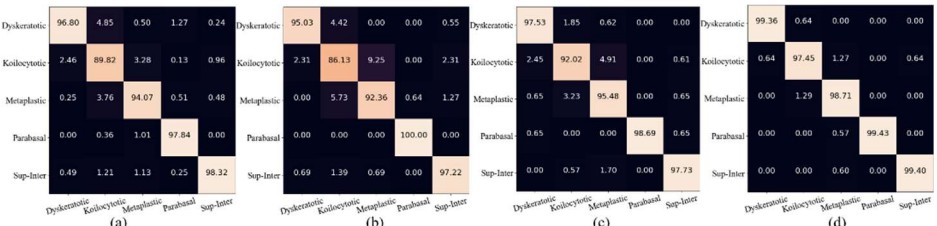

**Fig. 4.** Confusion matrices for classification on SIPaKMeD dataset: (a) CNN features, (b) ResNet-101, (c) DenseNet-121, (d) Our method.

**Results on Motic dataset.** In this experiment, the number of clustering and the neighbor parameter for Subset-1 are respectively determined as 2048 and 10 by experiments. For Subset-2, the number of clustering and the neighbor parameter are 2048 and 40. Note that the 27-dimensional cell morphological feature descriptor and SVM are also used for comparison. As presented in Table 4, CNN-based methods are significantly better than the method based on hand-crafted features (i.e. Morphological-27) in Subset-1 and Subset-2. Moreover, our method is superior to ResNet-101 and DenseNet-121 due to the reason that the latent correlations of images are effectively incorporated into CNN features via GCN. As illustrated in Fig. 5, the classification performance of our method for Koilocytotic cells outperforms other methods. For another two abnormal cells (i.e. atypical squamous cells and cells with High-N/C-Ratio), our method has more excellent classification results.

**Table 4.** Comparison of classification accuracies (mean±std) on Motic dataset (%).

| Methods | Morphological-27 | ResNet-101 | DenseNet-121 | Our method |
|---|---|---|---|---|
| Subset-1 | 84.35±1.02 | 89.39±0.74 | 92.13±0.74 | 94.90±0.25 |
| Subset-2 | 82.70±0.67 | 93.08±0.21 | 93.92±0.16 | 94.86±0.34 |

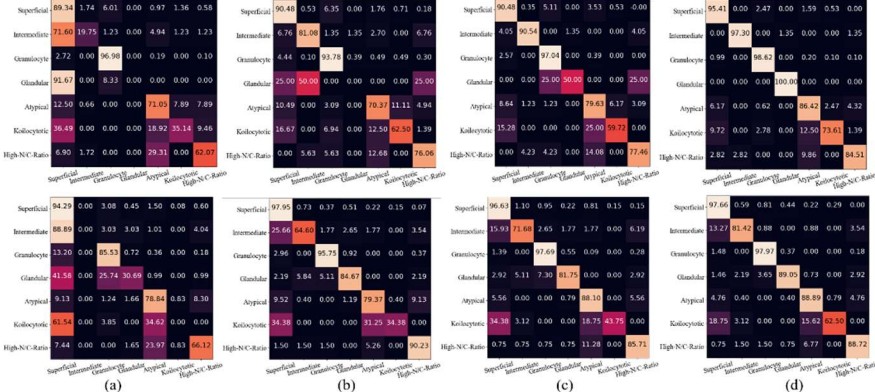

**Fig. 5.** Confusion matrices for classification on Motic Subset-1 and Subset-2: (a) Morphological-27, (b) ResNet-101, (c) DenseNet-121, (d) Our method. The first row is from Subset 1 and the last one is Subset 2.

## 4    Conclusion

In this paper, a novel cervical cell classification method based on Graph Convolutional Network (GCN) is developed and the large-scale Motic liquid-based cytology image dataset is also proposed. Our method uses the intrinsic clustering relationship to construct the graph structure and then generates the relation-aware representation through GCN, which is finally encoded into CNN features for improving classification performance. It not only effectively takes into account the potential correlation of images, but

also yields more discriminative feature representation. Experiments on cervical cell classification demonstrate the proposed method has better classification results.

## Acknowledgement

This work was supported by the National Natural Science Foundation of China (No. 61906058, 61771031 and 61901018), the Anhui Provincial Natural Science Foundation (No. 1908085MF210), and the Fundamental Research Funds for the Central Universities.

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
