# OpenReview forum: "Graph Convolutional Networks for Cervical Cell Classification"
_MICCAI.org/2019/Workshop/COMPAY — COMPAY 2019_

### Official Review · AnonReviewer3 · 2019-07-26
**Graph convolutional networks for cervical cell classification.**

**Rating:** 6
**Confidence:** 5

**Review:**

Summary:
The authors trained a system to classify cell images into different types combining CNN and GCN features. First, they pretrain the CNN to be able to extract an embedding vector for each cell image. Second, they apply k-means to these embeddings to find the most representative centroids, and use their intrinsic similarities to create a graph of cluster centroid correlation. Third, they run a GCN on top of this graph in order to integrate centroids’ neighbor information and find relation-aware features. Finally, they train the cell classifier (pretrained CNN and GCN from scratch) using the cross-entropy loss on cell types. The authors conduct a series of experiments to show that the use of the GCN features boosts classification performance compared to handcrafter and non-GCN methods.

Strong points:
* The paper is clearly written: the problem is nicely stated and the method extensively explained.
* They obtain excellent results.
* The proposed method is an interesting application of GCNs to images.

Constructive criticism:
* It is not entirely clear how the authors tuned several crucial hyper-parameters such as the k in k-means, the neighbor parameter, and the dimensions of the GCN layers. It seems that they chose them during cross-validation, which wouldn’t be correct if they report results on this same cross-validation sets. I would say there are three options: a) create a small validation set to tune these parameters separately, then perform the cross-validation on the remaining data to measure the final performance independently; b) divide the data into validation and test sets, use the validation set to tune the parameters via cross-validation, and the independent set to measure performance; and c) divide the data into N partitions, use N-2 for training, 1 for parameter tuning, 1 for independent testing, and rotate the partitions until you have measured performance independently in all partitions. In the current design, if they tune the parameters and measure performance on the same set, the results could be optimistically biased.
* I would like to understand what kind of information is driving the performance improvement. Maybe the authors could show images of samples that were misclassified without GCN features, but correctly classified when GCN features were included. Maybe there is a visual pattern that can be identified for interpretation purposes. It is difficult to understand why the relationship between images matters so much when classifying them.
* Not sure that the sentence “service provider of world-leading digital pathology and microscope solutions” is appropriate in a scientific paper.

---

### Official Review · AnonReviewer1 · 2019-08-05
**Strong results, some concerns**

**Rating:** 6
**Confidence:** 3

**Review:**

The paper tackles cervical cell classification with a framework combining a regular CNN with a graph convolutional network (GCN), capturing adjacency in the CNN feature space.

The strong point is clearly the results achieved, substantially better than the alternatives tested. It is also nice to see that a new dataset has been developed. However, it is unclear if and how this dataset is made accessible to the research community – which is needed to make it fully valuable.

As the GCN is the component causing the performance boost, it is particularly interesting for the reader to understand it in detail. For readers with limited experience of GCNs, such as this reviewer, there are a few unclarities. What is the GCN really learning, and why does it provide extra value? The relation awareness is repeatedly mentioned, but more concreteness is needed, such as giving examples and/or providing an intuitive idea of the benefits.

Other than the GCN learning, the paper is clear and understandable. The confusion matrices are, however, so small they are difficult to read.

The parameter selection for the proposed method is a bit concerning. The empirical approach for finding the best values could have compromised the training/validation setup if done on the same data, but it’s hard to tell from the text. The fact that different values have been chosen for all three datasets is a weakness, and causes concern about the sensitivity to that selection. What were the results for other parameter choices? Are the results better only for the best parameter choice? This should be discussed.

The statement “world-leading digital pathology and microscope solutions” is inappropriate in a scientific paper and should be removed. In my opinion, even if the results are strong, I would also use less bold statements than “excellent superiority” said in the abstract.

---

### Official Review · AnonReviewer2 · 2019-08-08
**Graph Convolutional Networks for Cervical Cell Classification**

**Rating:** 6
**Confidence:** 3

**Review:**

The paper demonstrates the use of a graph convolutional network for classification of cervical cell images into multiple classes. It also introduces a new dataset with gold standard. The method is evaluated both on a known public dataset as well as on the newly introduced one and compared against various other approaches.

The performance of the graph approach is good, in particular for some cell types that are easily confused. The output for such classes is much better.
There are, however, two aspects of the paper that prevent me from assigning a high score to this paper. First of all, I find the novelty of the paper limited. It is basically applying an existing technique to a different problem. There are some minor changes to the approach, but nothing substantial. Secondly, the description of the method is rather concise, making it hard to truly evaluate the validity of the experiments and the fairness of the comparisons.

I very appreciate the introduction of a novel dataset with gold standard, but the blatant advertising should be removed from this paper.

Overall, a potentially interesting application and a possible source of good discussions, but presented in a weak paper.

---

### Decision · Program_Chairs · 2019-08-20

Accept